Trans-species polymorphism at antimicrobial innate immunity cathelicidin genes of Atlantic cod and related species

Halldórsdóttir Katrín katrinhalldorsdottir@gmail.com
Árnason Einar
Institute of Life and Environmental Sciences, University of Iceland , Reykjavík , Iceland
Kumar Abhishek
Electronic publication date: 2015 May 21
Publication date: 2015
Volume: 3
Electronic Location ID: e976
Received 2015 Mar 24; Accepted 2015 May 5
Copyright: © 2015 Halldórsdóttir and Árnason
Copyright year: 2015
Copyright holder: Halldórsdóttir and Árnason
License: This is an open access article distributed under the terms of the Creative Commons Attribution License, which permits unrestricted use, distribution, reproduction and adaptation in any medium and for any purpose provided that it is properly attributed. For attribution, the original author(s), title, publication source (PeerJ) and either DOI or URL of the article must be cited.
License URL: https://creativecommons.org/licenses/by/4.0/

Keywords: Atlantic cod, Innate immunity, Cathelicidin, Balancing selection, Trans-species polymorphism, Gadids

Funding: Icelandic Science Foundation grant of excellence 40303011 University of Iceland Research Fund Svala Árnadóttir private foundation Funding was provided by Icelandic Science Foundation grant of excellence (nr. 40303011), a University of Iceland Research Fund grant, a Svala Árnadóttir private foundation grant to Einar Árnason and a doctoral grant from the University of Iceland Research Fund to Katrín Halldórsdóttir. The funders had no role in study design, data collection and analysis, decision to publish, or preparation of the manuscript.

==============================
Natural selection, the most important force in evolution, comes in three forms. Negative purifying selection removes deleterious variation and maintains adaptations. Positive directional selection fixes beneficial variants, producing new adaptations. Balancing selection maintains variation in a population. Important mechanisms of balancing selection include heterozygote advantage, frequency-dependent advantage of rarity, and local and fluctuating episodic selection. A rare pathogen gains an advantage because host defenses are predominantly effective against prevalent types. Similarly, a rare immune variant gives its host an advantage because the prevalent pathogens cannot escape the host’s apostatic defense. Due to the stochastic nature of evolution, neutral variation may accumulate on genealogical branches, but trans-species polymorphisms are rare under neutrality and are strong evidence for balancing selection. Balanced polymorphism maintains diversity at the major histocompatibility complex (MHC) in vertebrates. The Atlantic cod is missing genes for both MHC-II and CD4, vital parts of the adaptive immune system. Nevertheless, cod are healthy in their ecological niche, maintaining large populations that support major commercial fisheries. Innate immunity is of interest from an evolutionary perspective, particularly in taxa lacking adaptive immunity. Here, we analyze extensive amino acid and nucleotide polymorphisms of the cathelicidin gene family in Atlantic cod and closely related taxa. There are three major clusters, Cath1, Cath2, and Cath3, that we consider to be paralogous genes. There is extensive nucleotide and amino acid allelic variation between and within clusters. The major feature of the results is that the variation clusters by alleles and not by species in phylogenetic trees and discriminant analysis of principal components. Variation within the three groups shows trans-species polymorphism that is older than speciation and that is suggestive of balancing selection maintaining the variation. Using Bayesian and likelihood methods positive and negative selection is evident at sites in the conserved part of the genes and, to a larger extent, in the active part which also shows episodic diversifying selection, further supporting the argument for balancing selection.

Introduction

Vertebrates fight microbial infections using both innate immunity and adaptive responses. MHC molecules, cell surface molecules with broad (MHC-I) and specialized (MHC-II) pathogen recognition features (Murphy, Travers & Walport, 2007), show trans-species polymorphisms, variation indicative of adaptive balancing selection. For example, certain MHC-II alleles of humans are more closely related to certain alleles of chimpanzee than to other human alleles (Fan et al., 1989; Nei & Hughes, 1991). An ancient balanced polymorphism will generate long genealogical branches. Neutral variation will accumulate at sites close to the balanced polymorphic sites (Charlesworth, 2006). However, depending on recombination, the size of the genomic region can be quite short, making trans-species polymorphism hard to detect. Obvious and pervasive trans-species polymorphism, in contrast, is most likely due either to multiple sites under balancing selection or to suppression of recombination or to both (Wiuf et al., 2004). The models that have been proposed for detecting balancing selection in molecular data frequently assume that there is a single site under balancing selection. The silent and non-coding polymorphisms surrounding that site are taken as a signature of selection (Gao, Przeworski & Sella, 2015; Leffler et al., 2013). With the wealth of genomic data currently being generated, it is evident that many selective effects are related to immune defenses (Nielsen et al., 2007; Quintana-Murci & Clark, 2013; Teixeira et al., 2014; Osborne et al., 2013). Our understanding of balancing selection will be much improved by these new data, and important insights will be gained from genetic data without embarking on functional studies (Charlesworth, 2006).

Unique among vertebrates, the Atlantic cod (Gadus morhua) genome reveals the evolutionary loss of MHC-II and CD4, major parts of the adaptive immune system (probably they also are lost in other gadids, Star et al., 2011). Yet cod are healthy, playing a major ecological role in the North Atlantic, and are capable of sustaining large commercial fisheries. However, the way in which cod compensate for the lack of an adaptive immune response is unknown (Pilström, Warr & Strömberg, 2005; Magnadottir, 2010; Star & Jentoft, 2012). Host and parasite/pathogen interactions are very interesting in evolutionary terms. Pathogens set selective pressures on hosts and the response of the host is crucial for its own survival as well as the survival of the parasite. The innate immune system is at the forefront of this battle. It is of special interest to investigate evolution and variation of the innate immunity genes responsible for host defense.

Various families of antimicrobial peptides are an essential part of innate immunity. The cathelicidin family, first described in various mammals (Zanetti, Gennaro & Romeo, 1995), has been extensively studied in many organisms, e.g., primates (Zelezetsky et al., 2006) and fish (Maier et al., 2008; Kapralova et al., 2013). Important tools, such as Clnp-/- knockout mice, are available for functional studies (see e.g., Zhang et al., 2012). The number of genes coding for this protein varies among species. For example, there is a single gene in human (Gudmundsson et al., 1996) whereas there are ten in pig (Dawson et al., 2013). The protein is characterized by an N-terminus, a signal sequence, a conserved cathelin-like domain (exons 1, 2 and 3) and a C-terminal domain with antimicrobial activity (exon 4). The N-terminus of the protein has certain conserved features that characterize all cathelicidins, i.e., four cysteine residues forming two disulfide bridges (Tomasinsig & Zanetti, 2005) (Fig. S1). This evolutionarily conserved part is, nevertheless, targeted by positive selection (Zhu, 2008) (Fig. S1). The C-terminus is highly variable within multigene families and among species, most likely due to diversifying balancing selection (Tomasinsig & Zanetti, 2005). Many innate immune molecules have been described in Atlantic cod, e.g., piscidin (Fernandes, Ruangsri & Kiron, 2010), beta-defensin (Ruangsri et al., 2013) and the expanded toll-like receptor family (Sundaram et al., 2012), showing novel forms and patterns indicating importance of antimicrobial peptides and their genes for the immunity of these fish.

Several hypotheses have been proposed for the selective maintenance of high diversity at the MHC-II loci in vertebrates. These hypotheses include the heterozygote advantage hypothesis, the frequency-dependent rare-allele advantage hypothesis, and the fluctuating selection hypothesis under which the intensity of selective pressure can vary in accordance with the stimulus from pathogens. Thus, pathogen-driven episodic selection may vary in different environments and at different time periods (Clarke, 1962; Spurgin & Richardson, 2010; Sommer, 2005). However, the molecular signatures behind such balancing selection can be hard to detect and distinguish from other types of selection (Quintana-Murci & Clark, 2013).

Another example of unusually high polymorphism are the disease resistance R genes in Arabidopsis (Bakker et al., 2006). The mechanism behind extremely high gene copy number has been explained by the advantage of fixed heterozygosity based on duplicated genes each carrying different variants. This would give the advantage of overdominance without incurring any segregation load. In another study on R genes in the Arabidopsis, Shen et al. (2006) showed the effect of balancing selection in evolution of presence/absence polymorphism. In their study the R genes show different allele frequencies reflecting frequency-dependent selection at different stages of the evolutionary process.

Most genome-wide studies, scanning for variation, show high-frequency polymorphisms in genes related to immunity (Nielsen et al., 2007; Leffler et al., 2013). In this study, we examine the Cathelicidin family of innate immunity genes in Atlantic cod in individuals from throughout the distributional range (Fig. 1), and in closely related species. We report large variation within and among species. We report a distinctive data set discovered when we attempted to amplify a particular Cathelicidin gene with a pair of primers designed from Atlantic cod sequences. Our initial aim was to study population variation at the single codCath1 locus previously described (Maier et al., 2008) and also found in the Atlantic cod genome sequence (Star et al., 2011). With only a single pair of primers we found extreme variation in 97 clones from 27 individuals. The amount and patterns of variation both within and among species cannot be explained as single locus variation. We discuss paralogous variation and the orthologous variation within paralogs in terms of trans-species polymorphism.

Figure 1 Map of sampling sites of Atlantic cod and closely related species.

Locality codes for Atlantic cod samples are Can for Newfoundland, Canada, Gre for Greenland, Ice for Iceland, Nor for Norway, Bar for Barents Sea, Far for Faeroe Islands, Bal for Baltic Sea, and Cel for Celtic Sea. Species codes for closely related species are Gch for Gadus chalcogrammus and Gma for Gadus macrocephalus from the Pacific ocean (Pac), and Gog for Gadus ogac and Bsa for Boreogadus saida from Arctic Ocean in Greenland.

Materials and Methods

Sampling

We used 97 clones from 27 individuals in the study. We isolated DNA from gill filament tissue for samples from Iceland and from fin clips tissue for all other specimens. There were 19 individuals of Atlantic cod (mnemonic: Gmo) from throughout the distributional range of the species: two each from Greenland (Gre), Barents Sea (Bar), Celtic Sea (Cel), Baltic Sea (Bal), Norway (Nor), Faroe Islands (Far), and Canada (Can) and five from around Iceland (Ice). We randomly sampled the individuals from our large sample collection (Árnason & Halldórsdóttir, 2015) containing thousands of samples so as to cover a wide geographic area. We also included two individuals of each of the closely related species (Fig. 1) the Pacific cod Gadus macrocephalus (Gma), Greenland cod Gadus ogac (Gog), Walleye pollock Gadus chalcogrammus (Gch), and Polar cod Boreogadus saida (Bsa), which is more distantly related. Pacific cod is considered a speciation from an Atlantic cod invasion into the Pacific (Pac) at approximately 4 mya based on genomic mtDNA data, Greenland cod is a recent re-invasion of Pacific cod into the Arctic and Atlantic oceans, and Walleye pollock is a speciation from an Atlantic cod invasion into the Pacific at 3.8 mya (Coulson et al., 2006) (and see Carr et al., 1999; Pogson & Mesa, 2004). Labeling is as follows: Individuals are labeled with a six digit barcode, clones with a dash and a one or two digit clone number, species is labeled with the species mnemonic, and locality with the locality mnemonic.

Figure 2 Maximum likelihood phylogenetic tree of exon 4 with bootstrap values.

Phylogenetic tree built on amino acid sequences of exon 4, the active peptide in cathelicidin, from 43 clones of various individuals of Atlantic cod and four sister taxa. Bsa.Gre (Boreogadus saida), Gch.Pac (Gadus chalcogrammus), Gma.Pac (Gadus macrocephalus), Gog.Gre (Gadus ogac) and Gmo (Gadus morhua) from various locations: Iceland (Gmo.Ice), Greenland (Gmo.Gre), Barents Sea (Gmo.Bar), Celtic Sea (Gmo.Cel), Baltic Sea (Gmo.Bal), Norway (Gmo.Nor), Faeroe Islands (Gmo.Far), Canada (Gmo.Can).

The Icelandic Committee for Welfare of Experimental Animals, Chief Veterinary Office at the Ministry of Agriculture, Reykjavik, Iceland has determined that the research conducted here is not subject to the laws concerning the Welfare of Experimental Animals. (The Icelandic Law on Animal Protection, Law 15/1994, last updated with Law 157/2012.) DNA was isolated from tissue taken from dead fish on board research vessels. Fish were collected during the yearly surveys of the Icelandic Marine Research Institute. All research plans and sampling of fish, including the ones for the current project, have been evaluated and approved by the Marine Research Institute Board of Directors. The Board comprises the Director General, Deputy Directors for Science and Finance and heads of the Marine Environment Section, the Marine Resources Section, and the Fisheries Advisory Section. Samples were also obtained from dead fish from marine research institutes in Norway, the Netherlands, Canada and the US that were similarly approved by the respective ethics boards. The samples from the US used in this study have been described in Cunningham et al. (2009) and the samples from Norway in Árnason & Pálsson (1996). The samples from Canada consisted of DNA isolated from the samples described in Pogson (2001). The samples from the Netherlands were obtained from the Beam-Trawl-Survey (http://www.wageningenur.nl/en/Expertise-Services/Research-Institutes/imares/Weblogs/Beam-Trawl-Survey.htm) of the Institute for Marine Resources & Ecosystem Studies (IMARES), Wageningen University, the Netherlands, which is approved by the IMARES Animal Care Committee and IMARES Board of Directors.

Molecular analysis

We extracted genomic DNA using a Chelex/proteinase K extraction method (Walsh, Metzger & Higuchi, 1991). PCR was performed using Long PCR Enzyme Mix (Thermo Scientific/Fermentas #K0181) according to the manufacturer’s two-step cycling protocol. The PCR program was as follow: initial denaturation step of 3 min at 94 °C; 10 cycles of 20 s denaturation at 94 °C, 30 s annealing at 60 °C and 1.5 min extension at 68 °C. The annealing temperature was reduced by 2 °C in next two cycles and 1 °C in the following cycle, reaching annealing temperature of 55 °C. Additional 22 cycles were run under this condition with a 7 min final extension. Total 35 cycles.

The primers used for PCR were CodCathF1: 5′-TGTTCAGCACAAAGCCAAACT-3′ from Maier et al. (2008) and CodCathR4: 5′-GAGACAGGCTCAAGCCAATG-3′ a new reverse primer made for this study. The CodCathR4 primer was designed using Primer3 (Untergasser et al., 2012) and 3′ UTR part of GenBank sequence with accession number EU707291.1 as template.

Universal M13F and M13R primers were used for sequencing, using the BigDye® Terminator v3.1 Cycle Sequencing Kit (Applied Biosystems) according to the manufacturer’s protocols for plasmid sequencing except that we used 1/16 of the manufacturer’s recommended amount.

The PCR amplification fragments were gel extracted and cloned with PCR® 4-TOPO vector (Invitrogen™) and Sanger sequenced using an AB-3500xL Genetic Analyser (Applied Biosystems) (Halldórsdóttir & Árnason, 2009). All sequences were analyzed using the Phred/Phrap/Consed software suite (Ewing et al., 1998; Ewing & Green, 1998; Gordon, Abajian & Green, 1998) and had top-quality Phred score values (>30). Our initial goal was to sequence three clones from each individual to eliminate PCR errors according to a strategy that we discuss below and in Árnason & Halldórsdóttir (2015). The amplified fragment contained the whole gene, four exons and three introns with part of the 5′ and 3′ UTR (Fig. S2). We sequenced the gene and the 3′ UTR. EcoR1 digest of the clones run on agarose gels showed different sizes of the fragments in clones from some individuals. The size differences were confirmed upon sequencing. Therefore, we added and sequenced more clones from chosen individuals to further study the different sized fragments (see Table 1).

Table 1 Number of clones and number of forms or alleles in clones from different individuals.

Individuals are labeled by species and sampling locality. Individuals showing three different forms or alleles are marked with **.

nr	Barcode	Origin	Number of clones sequenced	Number of forms or alleles		
Atlantic cod	
1	105746	Gmo.Gre	3	2		
2	104931	Gmo.Gre	3	2		
3	140254	Gmo.Bar	3	1		
4	140272	Gmo.Bar	8	3	**	
5	118507	Gmo.Ice	12	3	**	
6	125968	Gmo.Ice	3	2		
7	118214	Gmo.Ice	3	1		
8	117795	Gmo.Ice	3	1		
9	117757	Gmo.Ice	3	1		
10	140179	Gmo.Cel	3	2		
11	140176	Gmo.Cel	3	1		
12	140219	Gmo.Bal	3	1		
13	140233	Gmo.Bal	3	1		
14	152921	Gmo.Nor	3	1		
15	152924	Gmo.Nor	3	1		
16	115574	Gmo.Far	2	2		
17	114718	Gmo.Far	6	2		
18	200093	Gmo.Can	6	2		
19	200079	Gmo.Can	3	2		
Closely related species	
20	103659	Bsa.Gre	3	1		
21	104725	Bsa.Gre	2	1		
22	103852	Gog.Gre	3	1		
23	104947	Gog.Gre	3	1		
24	152074	Gma.Pac	3	2		
25	152050	Gma.Pac	3	2		
26	152018	Gch.Pac	3	1		
27	152027	Gch.Pac	3	3	**	
	27	12	97			

Data analysis

Errors occur during PCR amplification and inevitably will be found, mostly as singletons, in the sequences of the cloned DNA. To remove this source of variation from the data we initially had planned to use the strategy of Árnason & Halldórsdóttir (2015) to get a consensus sequence for each individual from its three clones. However, the results showed that sequences of clones from some individuals were very different from each other, too divergent to be variation due to PCR errors. In some instances they belonged on the amino acid level to already described paralogous genes (Maier et al., 2008). Therefore, we revised the strategy for eliminating PCR errors by screening out singleton sites as follows. The three clones from each of the 27 individuals yielded 81 clones and, as already stated, we added extra clones for some individuals for a total of 97 clones. Singleton sites among the various clones from each individual that belonged to a certain cluster were considered PCR errors and not counted if that site was not found variable in clones from another individual (or other individuals). However, a singleton variant among the clones of an individual was considered a real SNP and was retained if that site was similarly variable in clones from other individuals (see Halldórsdóttir & Árnason, 2009, for estimation of errors in replicate PCR reactions). If a single clone from a particular individual represented a different cluster (paralogous gene) from the rest of the clones from that individual, then that clone was included in the analysis. If the same form was present in all clones from an individual only one sequence was included in the analysis (a consensus sequence for that individual). Using this strategy we had 43 clones. Each singleton site in the data of the 43 clones analyzed here was considered a real variant because it was found in more than one clone in the original data of 97 clones. The 43 clones analyzed here contain a single representative clone from each individual for either each allele or each gene. We also present an analysis of the 97 clones for comparison. New sequences generated in this study have GenBank accession numbers KJ831349–KJ831391.

EST sequences from the Canadian Atlantic Cod Genomics and Broodstock Development project (Bowman et al., 2011) were used in the analysis for comparison on the protein level. These were GenBank Accession numbers EY975127.1 (based on mRNA from a Gadus morhua spleen SSH library enriched for genes up-regulated by formalin-killed atypical Aeromonas salmonicida), FG312333.1 (based on Gadus morhua blood library injected with polyriboinosinic polyribocytidylic acid and formalin-killed Aeromonas salmonicida), and ES786338.1 (Gadus morhua spleen SSH library enriched for genes up-regulated by polyriboinosinic polyribocytidylic acid). We also used GW862872.1 (based on mRNA from thymus from a Norwegian coastal cod, already charcaterized as cod Cathelicidin 2 in Maier et al., 2008) and EU707291.1 (complete cds from mRNA isolated from a wild cought cod from Iceland, previously characterized as cod Cathelicidin 1, codcath1 in Maier et al., 2008). Finally, we also included the complete gene sequence from GeneScaffold 2759 from the North East Arctic Atlantic cod genome sequence (Star et al., 2011) available on the Ensembl browser (Flicek et al., 2014).

Phylogenetic maximum likelihood trees were built using Muscle (Edgar, 2004) aligned sequences with a branch support of 100 bootstrap replicates using phyML (Guindon & Gascuel, 2003) through Seaview (Gouy, Guindon & Gascuel, 2010). Translations of our original nucleotide data were performed with EMBOSS Transeq (http://www.ebi.ac.uk/Tools/st/emboss_transeq/). We used DNAsp (Librado & Rozas, 2009) and R (R Core Team, 2014) and the ape, pegas, seqinr, ade4, adegenet and LDheatmap packages (Paradis, Claude & Strimmer, 2004; Paradis, 2010; Charif & Lobry, 2007; Dray & Dufour, 2007; Jombart & Ahmed, 2011; Shin et al., 2006) for population genetic and statistical analysis. We performed Discriminant Analysis of Principal Components (DAPC) with functions from the adegenet package. We used TexShade (Beitz, 2000) for visual presentation of alignments. We used ggtree (Yu, 2015, see https://github.com/GuangchuangYu/ggtree) based on ggplot2 (Wickham, 2009) to present secondary structure data.

For codon-based likelihood and Bayesian analysis of selected sites, we used the website www.datamonkey.org (Delport et al., 2010; Pond, Frost & Muse, 2005). The following methods were used to search for positively and negatively selected sites: MEME (Murrell et al., 2012), SLAC, FEL and REL (Kosakovsky Pond & Frost, 2005) and FUBAR (Murrell et al., 2013). Indels were excluded from the analysis of exon 4, and, therefore, only sites found in Cath2 that were common to the three genes were analyzed. The p-values were set at 0.2 for the SLAC, FEL and MEME programs to generate the results in Tables 2 and 3. The REL Bayes Factor was 50, and the FUBAR Posterior Probability was 0.9.

Table 2 Codon-based maximum likelihood and Bayesian analysis for positively selected sites in exon 4 and exons 1, 2, and 3 combined.

Statistics with significance level p < 0.05, posterior probability >0.9 and Bayes Factor >50 are boldfaced. Consensus column summarizes methods which found the codon positively selected with significance level p < 0.2. Analysis was made using the Datamonkey server www.datamonkey.org (Delport et al., 2010; Pond, Frost & Muse, 2005).

Codon	SLAC	SLAC	FEL	FEL	REL	REL	MEME	MEME	FUBAR	FUBAR	Consensus	
	dN − dS	p-value	dN − dS	p-value	dN − dS	Bayes F	ω +	p-value	dN − dS	Post.Pr.	S	F	R	M	Fu	
Positively selected sites in exon 4	
24	3.97	0.63	9.15	0.20	1.60	2.00	>100	0.13	0.40	0.83		+		+		
28	4.79	0.49	10.80	0.15	1.63	2.14	>100	0.19	0.40	0.83		+		+		
30	6.63	0.46	9.98	0.31	1.46	1.41	>100	0.02	0.41	0.87				+		
45	−3.49	0.92	−12.31	0.35	−0.45	0.01	>100	0.04	0.51	0.32				+		
47	4.92	0.46	7.18	0.21	1.61	2.21	>100	0.046	0.37	0.83				+		
48	4.61	0.52	8.18	0.30	1.46	1.58	>100	0.13	0.30	0.81				+		
49	5.50	0.41	13.24	0.12	1.65	2.23	>100	0.16	0.55	0.86		+		+		
51	6.90	0.37	12.22	0.10	1.71	2.36	>100	0.11	0.84	0.94		+		+	+	
55	4.79	0.48	10.78	0.13	1.68	2.45	>100	0.03	0.42	0.84		+		+		
57	7.27	0.32	12.63	0.11	1.68	2.21	>100	0.14	0.74	0.92		+		+	+	
59	4.94	0.45	8.24	0.19	1.63	2.23	>100	0.22	0.29	0.81		+				
62	0.43	0.72	4.43	0.72	−0.01	0.02	>100	0.01	0.40	0.66				+		
Positively selected sites in exons 1, 2, and 3 combined	
42	10.51	0.38	240.59	0.11	7.67	17.23	>100	0.14	4.37	0.97		+		+	+	
89	8.90	0.61	189.86	0.29	7.25	10.16	>100	0.30	3.43	0.93					+	

Table 3 Codon-based maximum likelihood and Bayesian analysis for negatively selected sites in exon 4 and in exons 1, 2, and 3 combined.

Statistics with significance level p < 0.05, posterior probability >0.9 and Bayes Factor >50 are boldfaced. Consensus column summarizes methods which found the codon positively selected with significance level p < 0.2. Analysis was made using the Datamonkey server www.datamonkey.org (Delport et al., 2010; Pond, Frost & Muse, 2005).

Codon	SLAC	SLAC	FEL	FEL	REL	REL	FUBAR	FUBAR	Consensus	
	dN − dS	p-value	dN − dS	p-value	dN − dS	Bayes F	dN − dS	Post.Pr.	S	F	R	Fu	
Negatively selected sites in exon 4	
2	−5.63	0.31	−15.55	0.08	−1.41	513.1	−0.63	0.79		–	–		
7	−3.09	0.51	−5.92	0.58	−0.38	147.9	−0.133	0.51			–		
13	−2.35	0.57	−2.43	0.81	−0.36	10777.7	−0.208	0.53			–		
14	−12.35	0.14	−55.82	0.02	−0.67	16734.7	−3.53	0.96	–	–	–	–	
31	−2.26	0.58	−2.17	0.83	−0.35	13704.8	−0.146	0.51			–		
45	−3.49	0.50	−12.31	0.35	−0.45	102.0	−0.509	0.62			–		
46	−9.93	0.12	−17.62	0.06	−1.28	2384580.0	−1.53	0.94	–	–	–	–	
58	−5.88	0.37	−8.66	0.36	−0.65	197.32	−0.62	0.66			–		
60	−2.18	0.56	−0.63	0.97	−0.02	112811000.0	−0.03	0.34			–		
62	0.43	0.73	4.43	0.72	−0.01	52.6	0.40	0.25			–		
68	0.04	0.71	1.10	0.92	−0.08	13381.4	0.09	0.32			–		
Negatively selected sites in exons 1, 2, and 3 combined	
15	−11.59	0.24	−210.0	0.10	−7.08	338.4	−4.50	0.97		–	–	–	
17	−15.07	0.11	−345.8	0.02	−7.15	579.5	−5.65	0.98	–	–	–	–	
26	−8.34	0.39	−128.4	0.45	−6.38	68.0	−2.93	0.73			–		
47	−7.54	0.33	−131.9	0.15	−7.1	501.4	−2.06	0.83		–	–		

Due to the trans-species nature of variation, some analysis that are developed for intraspecific variation were made on the trans-species variation. The assumption here is that trans-species variation is representative of intraspecific variation that could be found with larger sample sizes within each species.

Results and Discussion

Clusters of Cathelicidin variation

The variation clusters by tree building into three major groups (Fig. 2) that we call Cath1, Cath2, and Cath3. Cath1 has already been described as a single gene and characterized by Maier et al. (2008); Cath2, was originally described by Maier et al. (2008) based on a partial sequence from Canadian cDNA databank, and which we fully sequenced here. Cath2 was described as a paralogue of Cath1 (Maier et al., 2008). The third major group, Cath3, was novel and has not been described before. Only one of these genes, Cath1, was found in the cod genome assembly (www.ensemble.org, Star et al., 2011). However, the Atlantic cod genome sequence is incomplete with 611 Mb of 830 Mb assembled into scaffolds (Star et al., 2011) and probably is missing genes (Zhuang et al., 2012). Maier et al. (2008) had named a variant, for which they had found a cDNA sequence in GenBank, and that was characterized relative to Cath1 by a 10 amino acid indel, as Cath3. We found the same variant (117757_1.Gmo.Ice, Figs. 2 and 3) in our data as a variant of Cath1. Therefore, we drop the Cath3 label for this variant of Cath1 and henceforth use Cath3 for one of the major clusters of Figs. 2 and 3.

Figure 3 Maximum likelihood phylogenetic tree of all clones with bootstrap values.

Phylogenetic tree built on nucleotide sequences found in 97 clones from various individuals of Atlantic cod and four closely related taxa. Bsa.Gre (Boreogadus saida), Gch.Pac (Gadus chalcogrammus), Gma.Pac (Gadus macrocephalus), Gog.Gre (Gadus ogac) and Gmo (Gadus morhua) from various locations: Iceland (Gmo.Ice), Greenland (Gmo.Gre), Barents Sea (Gmo.Bar), Celtic Sea (Gmo.Cel), Baltic Sea (Gmo.Bal), Norway (Gmo.Nor), Faeroe Islands (Gmo.Far), Canada (Gmo.Can).

Orthologs and paralogs

An obvious question is whether these clusters represent orthologous or paralogous genes and alleles. Cath1 and Cath2 have already been established as paralogs (Maier et al., 2008). In our data clones from individual 118507.Gmo.Ice belonged to all three major clusters, Cath1, Cath2, and Cath3 (Fig. 2). Allelic variation at a single locus would only yield two forms in a diploid organism. Therefore, the three clusters must represent at least two paralogous genes. Similarly clones from Walleye pollock individual 152027.Gch.Pac also belonged to the three clusters (Fig. 2). Cath2 was most divergent. The Cath2 sequences, individuals in row 9–16 in Fig. 4 and Fig. S2, were considerably shorter than both Cath1 and Cath3 sequences or about 1210 bp long compared to about 1310–1368 bp (and see discussion on length variation below). Individual variation was found in a repeats at the beginning of intron 3 and an indel in exon 4 in Atlantic cod from Celtic sea (individual 140179.Gmo.Cel). Compared to the other two groups Cath2 had deletions in intron 3 and exon 4 (Fig. S2). The amino acids sequence in exon 4, the active peptide, also were different from the two other groups (Fig. 4). Thus, we consider Cath2 to be paralogous to the other clusters in accordance with Maier et al. (2008).

Figure 4 Alignment of exon 4, the major peptide in cathelicidin, from various individuals of Atlantic cod and four closely related taxa.

The sequences are grouped in accordance with the clades shown in Fig. 2. The first two groups are Cath3, the third group is Cath2, and the last group represents Cath1. Bsa.Gre (Boreogadus saida), Gch.Pac (Gadus chalcogrammus), Gma.Pac (Gadus macrocephalus), Gog.Gre (Gadus ogac) and Gmo (Gadus morhua) from various locations; Iceland (Gmo.Ice), Greenland (Gmo.Gre), Barents Sea (Gmo.Bar), Celtic Sea (Gmo.Cel), Baltic Sea (Gmo.Bal), Norway (Gmo.Nor), Faeroe Islands (Gmo.Far), Canada (Gmo.Can). Up arrows represent positively selected sites and down arrows negatively selected sites in Tables 2 and 3. (Fig. S1 shows the same for the conserved part in exons 1–3).

Furthermore, clones from individual 140272.Gmo.Bar belonged to both Cath2 and Cath3 (Fig. 2). Two Cath2 clones from this individual that differed by several sites, probably representing allelic variation at Cath2. This is further support that the Cath2 and Cath3 clusters represent paralogous genes. Clones from individual 140179.Gmo.Cel belonged to Cath2 and Cath3 (Fig. 2). The two Cath2 clones were identical and differed from the Cath2 of individual 140272 by several sites and an indel that is indicative of the variation among clones within the Cath2 cluster.

Clones from individual 104931.Gmo.Gre belonged to Cath1 and Cath2. There was only singleton variation, probably PCR error, between the two Cath2 clones. The Cath1 clone had very similar amino acid sequence to Cath1 clones from other individuals (Fig. 2) yet it differed somewhat at the nucleotide level (Fig. 3).

Clones from Pacific cod individual 152074.Gma.Pac belonged to both Cath1 and Cath3. If Cath1 and Cath3 are orthologous it would imply deeply divergent alleles at that locus. Similarly, clones from Pacific cod individual 152050.Gma.Pac belonged to both Cath1 and Cath3. The Cath3 clones (clones 1 and 2; Fig. 3) had identical amino acid sequence to clones from three other individuals: Greenland cod 103852.Gog.Gre, Atlantic cod 105746.Gmo.Gre and the other Pacific cod already mentioned 152074.Gma.Pac. At the nucleotide level the two Cath3 clones of 152050.Gma.Pac differed from each other by a few singleton sites that were probably due to PCR errors. It clustered with the other Pacific cod clones showing similar singleton variation at the nucleotide level (Fig. 3 and Fig. S2).

The sequences for different groups/alleles were of different sizes. The Cath3 cluster showed two subgroups or clades (A and B) that had some length differences. The first four clones in the alignment (Fig. 4 and alignment of the whole sequence in Fig. S2) are 1322 bp long except the clones of individual 152074.Gma.Pac which were 1237 bp long because of an indel in intron 3 and exon 4. The second subgroup or clade of Cath3 (the next four sequences in Fig. 4) were 1321, 1281, 1281 and 1276 bp long respectively due to length variation in intron 3 (Fig. S2). The Cath1 sequences, which constitute the rest of the sequences in Fig. 4, were from 1318 to 1368 bp long. Some variation was found in intron 3 (Fig. S2). For example, individual 152027.Gch.Pac had a long insertion but individuals 104947.Gog.Gre and 152050.Gma.Pac had deletions. Some minor variations were found in other individuals in intron 3, e.g., a repeats at the beginning of the intron. Individuals 104947.Gog.Gre and 152050.Gma.Pac had deletions in exon 4 but individuals 114718.Gmo.Far, 117757.Gmo.Ice, 105746.Gmo.Gre and 152074.Gma.Pac had insertions.

The three clusters probably represent functional genes. The cDNA sequences that we included are based on expressed sequences and they belonged to the Cath1 and Cath2 clusters. There were no signs of lack of function for Cath3.

From these considerations, we consider Cath2 to be a paralog of the Cath1 and Cath3 clusters. Based on the tree, the overall divergence between Cath1 and Cath3 was similar to the divergence of Cath2 from both Cath1 and Cath3 (Fig. 2). However, the sequence similarity is much higher between Cath1 and Cath3 than between Cath1 or Cath3 on one hand and Cath2 on the other, both at the nucleotide and amino acid levels (Fig. 4, Figs. S1 and S2). Cath1 and Cath3 probably are paralogs although we do not have conclusive evidence for that deduction. However, if they are orthologs it will strengthen our main hypothesis of trans-species level of variation. Furthermore, one could argue that the two Cath3 clades represented paralogous genes. If that were the case, it would also strengthen our hypothesis of trans-species polymorphism because variation within both (A and B) forms of Cath3 clusters by alleles and not by species. The discriminant analysis of principal components (DAPC) lends further support that the variation clusters by alleles (Fig. 5) and not by species (Fig. 6). The DAPC cleanly separated groups defined by alleles but groups based on species were largely overlapping. We thus conclude that there are three paralogous genes, Cath1, Cath2, and Cath3, and that the variation within each cluster represents allelic variation of each gene. The most important result is the trans-species nature of the variation in that each allele group contains representatives of various species.

Figure 5 Discriminant Analysis of Principle Components (DAPC) scatterplot of the five allele clusters.

Ten principle components and three discriminant functions were retained in the analysis. Scatterplot of the first two disciminant functions with eigenvalues used in black. The alleles are represented as dots of different shapes and colors representing the a priori groups Bsa (Boreogadus saida), and the Cath1, Cath2, Cath3-A and Cath3-B clusters of Fig. 2.

Figure 6 Discriminant Analysis of Principle Components (DAPC) scatterplot of the five species clusters.

Ten principle components and three discriminant functions were retained in the analysis. Scatterplot of the first two disciminant functions with eigenvalues used in black. The species are represented as dots of different shapes and colors representing the a priori groups of species: Bsa (Boreogadus saida), Gch Gadus chalcogrammus, Gma Gadus macrocephalus, Gmo Gadus morhua, and Gog Gadus ogac.

In some individuals we found representatives of only one gene or even of only a single allele. In some instances, we looked more closely at several clones of such individuals without detecting more alleles. This may be a chance event or it may be due to variation in primer binding sites. In that case, our data would have ascertainment bias from using only a single primer pair for PCR amplification. If that were the case, we are missing even more alleles. Similarly, a single Cathelicidin, Cath1, is found in the cod genome assembly (www.ensemble.org, Star et al., 2011) which may indicate a single gene in that individual. However, the incompleteness of the genome assembly also may explain that. A further exploration of the possibility of copy number variation is one avenue for further studies. For example, whole genome or targeted sequencing of individuals showing different forms of Cathelicidins could reveal if there is copy number variation. If so, selection might be on the level of gene number as is the case in presence/absence polymorphism in R genes in Arabidopsis (Shen et al., 2006). If a duplicated gene is being selected for or against, copy number variation may confound the detection of selection by the various methods we have used.

Trans-species polymorphic variation

The major feature of the results is that within each paralogue the clones cluster by alleles and not by species. This is the hallmark of a trans-species polymorphism (Leffler et al., 2012; Leffler et al., 2013; Eimes et al., 2015). We have found trans-species polymorphisms of the cathelicidin genes and their alleles of Atlantic cod and closely related taxa that are akin to the human vs. chimpanzee MHC-II (Fan et al., 1989). The same topology was found for trees based on amino acid sequences of exon 4, the active part (Fig. 2), the amino acid sequences of exons 1–3, the conserved part, and, based on the nucleotide sequences for the whole genes (Figs. S2 and S3) for the 43 clones used. The tree based on nucleotide sequences of the complete genes for all 97 clones (Fig. 3) also showed the three distinctive groups clustering by alleles and not by species as also seen in the DAPC results as already stated. Thus, the profuse nucleotide and amino acid variation within each of the three paralogous genes fell into distinct clades with forms or alleles of the closely related species intertwined (Figs. 2–6 and Figs. S1–S4).

Signatures of gene conversion

Although no recombination was found by GARD, and visual inspection did not show four gametes, the sequences showed signatures of gene conversion (Lamb, 1984; Chen et al., 2007) (Fig. S2).

For instance, the individual clone 152027-1.Gch.Pac (individual eight in the Cath1 group in Fig. 4) clusters within Cath1. However, the first two highlighted amino acids (aa) are the same as in Cath3. The third aa highlighted in this individual, aa 42 (S), resembled that found in Boreogadus saida (the most distantly related taxon) and aa 48 (K) is identical to that of Cath2 for 152018-3.Gch.Pac. That aa is therefore unique for the Gadus chalcogrammus (Gch) species.

The peptides of clones of individuals 105746-3.Gmo.Gre and 152074-3.Gma.Pac in the Cath1 group (first two individuals in the Cath1 group in Fig. 4) have an insertion of five aa after site 24; they have L in site 51, as found in Cath2, a unique I in position 61 and K in position 66. There was thus unique allele of Cath1 found in two different species a clear case of trans-species variation.

The peptides of clones of individuals 152050-3.Gma.Pac (Gadus macrocephalus) and 104947-2.Gog.Gre (Gadus ogac) (individuals three and four in Fig. 4) show the same gap (or deletion) as in Cath2 (between sites 32 and 45) and R in position 24, also found in Cath2 and Cath3, they share unique aa in sites 54 and 66 (S and K) but after that position they resemble Cath1. These patterns are indicative of gene conversion. In this case, we have two alleles in Cath1 that are found in different species. These alleles are more closely related to each other than to other alleles from the same species, i.e., again a trans-species level of variation.

The aa sequence AYSIN at the C-terminus of the peptide is characteristic of the second of the two alleles of Cath3 (B) in our data (individual four to eight in the alignment in Fig. 4; the other allele (A) was characterized by the similar sequence AYIIN). However, this aa sequence also is found in the EST sequence FG312333.1 from Canada (individual six in Cath1 group in Fig. 4), which is clearly a Cath1 sequence elsewhere. This again is indicative of gene conversion and an indication of trans-species level of variation.

The peptide of individual 117757-1.Gmo.Ice (individual 11 in Cath1 group in Fig. 4) has the nine aa insertion that previously had been described as a paralogous gene Cath3 (Maier et al., 2008). According to our data it is an allelic variant of Cath1. Therefore, we drop the Cath3 label for this variant and reserve that for the major cluster (Fig. 2). Interestingly a shorter insertion of six aa (similar but not identical) was also found in individual 114718-4.Gmo.Far, an Atlantic cod from the Faeroe Islands.

Population genetic statistics

We estimated the nucleotide diversity π, the scaled mutation rate θ and Tajima’s D in a sliding window of 100 bp over the genes coding for Cath1 and Cath3, noncoding regions and both synonymous and non-synonymous sites in coding regions. For Cath1, θ was higher than π, giving a negative D over the whole gene (Figs. S5 and S6) with a high peak in exon 4 implying either purifying selection or demographic population expansion. Negative Tajima’s D can also indicate a selective sweep of positive selection and at several sites D < − 2 was statistically significant. In contrast, for Cath3, π was generally larger than θ, giving a positive D for almost all sites, with high and significant peaks (D > 2) in exon 4 (Figs. S7 and S8). This implies balancing selection or demographic population subdivision and bottlenecks. There also was much variation in non-coding regions, predominantly in introns. Intronic variation in the distinct clusters were in linkage disequilibrium with the non-synonymous variation found in exon 4 (Fig. 4 and Fig. S2).

We estimated linkage disequilibrium D′ among highly polymorphic sites (with a minor allele frequency at least three sequences out of 36; Fig. 8 and two out of 22 in Fig. 7). We excluded low polymorhic sites for clarity. Cath1 alone showed linkage disequilibrium between sites in the active part (exon 4) and the conserved part (exon 1–3) and sites in intron 3 (Fig. 7). If we consider Cath1 and Cath3 as one orthologous gene and consider the variants from the various species simply as representative of allelic variation within any single species, we can estimate linkage disequilibrium among that group of clones (all alleles from Cath1 and the two Cath3 clusters independent of species Fig. 8). With these assumptions, we found even stronger linkage disequilibrium between sites in exon 4 and intron 3. Overall, this may indicate the presence of control sequences in intron 3. However, these overall summary statistics may miss important details of selection. Therefore, we decided to examine what a codon-based analysis, skipping intronic variation, might reveal about selection.

Figure 7 Linkage disequilibrium D′ heatmap of high frequency polymorphic sites for Cath1 in Atlantic cod only.

Minor allele frequency set at 2/22.

Figure 8 Linkage disequilibrium D′ heatmap of high frequency polymorphic sites for Cath1 and Cath3 combined from all species.

Minor allele frequency set at 3/36.

Codon based analysis

In order to screen for purifying or positive selection acting on the protein we used several routines in Datamonkey server: www.datamonkey.org (Delport et al., 2010; Pond, Frost & Muse, 2005). This server provides several methods for detecting various forms of selection (Tables 2 and 3). We screened alignments for recombination with GARD (Kosakovsky Pond et al., 2006) and found no sign of recombination.

We analyzed exons 1–3, the conserved part, separately from exon 4, which constitutes the active peptide. Sites containing gaps were excluded from this analysis. Therefore, the analysis was done only on sites found in all three groups. The analysis estimated synonymous (S) and non-synonymous (N) changes within each codon and calculated either the ratio dN/dS or the difference dN − dS. For the codons with significant results, described below, both dN and dS were greater than zero. We compared several methods, SLAC, REL, FEL, MEME and FUBAR (Kosakovsky Pond & Frost, 2005) to detect amino acid sites under selection (Tables 2 and 3).

The SLAC (Single Likelihood Ancestor Counting) program, the most conservative compared with the empirical Bayesian and likelihood approaches, found no evidence of selection. Similarly, FEL (Fixed Effects Likelihood), which is less conservative, found no evidence of selection. However, REL (Random Effects Likelihood) found no positively selected sites but found 11 and four negatively selected sites in exon 4 and exons 1–3, respectively. A REL Bayes factor higher than 10 is strong evidence of selection, giving support to positively selected sites in exons 1–3, as also found by FUBAR. REL is highly sensitive but has a tendency to produce false positives because of an a priori defined distribution of rates to be fitted; therefore, it may misinterpret a new distribution of rates (Kosakovsky Pond & Frost, 2005). FUBAR (Fast Unconstrained Bayesian AppRoximation, Murrell et al., 2013) uses MCMC to avoid constraints on the distribution of the selection parameter. For FUBAR we consider a posterior probability of 0.95 as a stringent cutoff, 0.90 as a strong cutoff, and 0.80 as a suggestive cutoff. FUBAR found two positively and two negatively selected sites both in exon 4 and in exons 1–3 using the strong cutoff. Ten of twelve sites (Table 2) have posterior probabilities (for ω = β/α > 1 at a site) higher than the suggestive cutoff 0.8 (more than six-fold higher than the expected number of false positives of 1.6 with CI [0–4]). MEME (Mixed Effects Model of Evolution Murrell et al., 2012) might be the most appropriate method for our data because this method detects selection varying across lineages and identifies episodic and pervasive positive selection. MEME detected five sites with p ≤ 0.05 indicative of selection (Table 2). It is noteworthy that sites that are significant by one method (MEME) are not significant by another method (FUBAR) (sites 51 and 57 and the other way around for site 45). MEME can identify diversifying evolution in a subset of branches, where more restricted methods identify only purifying selection. Examples of this situation are sites 45 and 62 (Fig. 4 and Table 2), positively selected with p < 0.05 by MEME but negatively selected by REL.

Overall, the results of the exploratory codon-based analysis are in line with the results of the summary statistics (π and Tajima’s D) indicating positive and balancing selection mainly in exon 4, the active part. Both results add support for the inference of balancing selection based on the trans-species nature of the within paralogs variation.

Secondary structure predictions

Given the support for diversifying selection it is worthwhile to ask if predictions of protein structure of the active peptide would add support for the role of selection. We used the RaptorX protein structure server (http://raptorx.uchicago.edu/, Källberg et al., 2012) to predict secondary structure of exon 4, the active peptide. This program can give some predictions of structure without the use of close homologs in the protein structure databases. Because of how diverse the peptides are, it is difficult to use more accurate programs like pymol which rely on close homology of the predicted and template proteins from protein structure databases.

The results of the analysis showed that most sequences were predicted as rod-like linear Glycine rich structures. In all three groups there were sequences which predicted α helical structures and among Cath3 sequences there also were predictions of beta-hairpin structures (Fig. 9). The sequence variation of the Glycine, Serine, and Arginine rich part of the peptide (Fig. 4) may be responsible for these differences in predicted structure.

Figure 9 Predicted secondary structures of peptides in each group on a maximum likelihood phylogenetic tree of amino acid sequence of exon 4.

Secondary structure predictions were made using the RaptorX protein structure server (http://raptorx.uchicago.edu/, Källberg et al., 2012.

The exact impact on the protein structure, of mutations between the highly different alleles, will not be described here. However, robust prediction of the secondary structures for the mature antimicrobial peptide part of the gene, show variation that may indicate different biological function of the proteins of these alleles to a variety of microbes (Fig. 9) (Tomasinsig & Zanetti, 2005; Zhu & Gao, 2009). The predicted peptides described here are highly cationic. Their size ranges from 50 to 81 amino acid residues. The more positively charged the peptides, the stronger they bind to bacterial membranes (Bals & Wilson, 2003). Most of the peptides have linear secondary structure which presumably prevent α-helical conformation as is known for Proline rich peptides (Tomasinsig & Zanetti, 2005).

In mammals there is at least one cathelicidin peptide with α-helical conformation. This peptide folds into an amphipathic helical structure in connection with biological membranes (Tomasinsig & Zanetti, 2005). The first Cathelicidin identified in fish was from the Atlantic hagfish, Mysine glutinosa, with the mature peptide showing α-helical conformation (Uzzell et al., 2003). Few or any other Cathelicidins in fish have so far been shown to adopt α-helical conformation. In our data, we have prediction of peptides in all three groups i.e., Cath1, Cath2 and Cath3, which adopt this α-helical structure. Broekman et al. (2011a) made developmental expression studies with antibody from the mature peptide of Cathelicidin 1 in Atlantic cod. They show that the peptide has broad activity against different stimuli (Broekman et al., 2011b). Interestingly, the antibody they use was raised against the 14 amino acids which do not differentiate the three groups that we describe here (Broekman et al., 2011b). Therefore, it will be of interest to test whether the different forms described here have different activities and whether that could explain the broad activity they found. These future studies of the activity of the different peptides, will also be very interesting in the context of the rising interest in fish antimicrobial peptides in clinical dermatology (Rakers et al., 2013) and therapeutic antimicrobials (Masso-Silva & Diamond, 2014).

Spatial population differentiation

There has been a long-standing debate about the possible population differentiation of Atlantic cod (Jónsdóttir et al., 1999; Árnason, 2004; Eiríksson & Árnason, 2013). The genes behind primary defense against pathogens, like cathelicidin, are presumably under strong selection. It is expected that such loci will show pattern of geographic subdivision in contrast to loci with genome wide effect which relay demographic effects. However, there is no particular geographic structure evident among localities by visual inspection. For example, three individuals of Atlantic cod from Faroes, Norway and Canada show one of the alleles found in Cath1 (three aa highlighted in individuals 115574-2.Gmo.Far 152924-2.Gmo.Nor 200093-5.Gmo.Can in Fig. 4). In general the different specific variants were widely dispersed as expected of allelic variation of an ancient polymorphism.

Balancing selection

The shared polymorphism within paralogs found in our data, e.g., between Atlantic cod and Walleye pollock, suggests long-lasting maintenance by balancing selection. A trans-species polymorphism is in general a most important indication of balancing selection (Charlesworth, 2006). With an approximately five-year generation time and an effective population size (Ne) of approximately 10.000 in Atlantic cod (Árnason, 2004), the approximately 4 mya divergence time between the species (Coulson et al., 2006) is 20Ne, or five times higher than the average 4Ne fixation time for neutral variation (Clark, 1997). Such long-lasting trans-species polymorphism is often thought to be indicative of balancing selection (Hughes, 2002; Sommer, 2005). These considerations are based on the time scale of the Kingman coalescent (Kingman, 1982). The faster time scales of the multiple-merger coalescent, which are more appropriate for the high fecundity Atlantic cod (Birkner, Blath & Eldon, 2013; Árnason & Halldórsdóttir, 2015), would make this even more significant.

We show that the polymorphism is older than speciation given that divergent alleles of different paralogous genes can be found in different species. The balancing selection hypothesis is a plausible explanation because a scenario of concerted evolution between paralogous genes would otherwise be expected (Liao, 1999).

Conclusion

Trans-species polymorphism is in general strong evidence for balancing selection. We found a highly variable polymorphism at antimicrobial Cathelicidin loci with trans-species level of variation that suggests maintenance by some form of balancing selection. Given the functional role of the cathelicidin peptides and the diverse structures predicted the system may play an important role in a host/pathogen arms race. This may imply that negative frequency dependent and possibly episodic selection may be responsible for the balancing selection.

Further experiments are needed to test the activity of various cathelicidin peptides against a variety of microbes to both elucidate the mechanisms of selection (Nielsen et al., 2007; Quintana-Murci & Clark, 2013) and to better understand the expression of the various genes in relation to microbial infection. Further intra- and interspecific experiments are also needed to find out if there are more paralogous genes in the genome (c.f. pigs Dawson et al., 2013) and to establish their paralogous and orthologous relationships. This should include population genetics studies using stringent experimental protocols to avoid PCR and cloning artifacts (c.f. Lenz & Becker, 2008).

Using a phylogenetic analysis Star & Jentoft (2012) show an expansion of MHC-I and various Toll like receptor genes coinciding with the loss of MHC-II (Star et al., 2011). Atlantic cod may thus compensate evolutionary for the loss of MHC-II. Our results imply evolutionary forces shaping variable innate immunity under selection pressure from contacts between hosts and microbes (Barreiro & Quintana-Murci, 2010; Quintana-Murci & Clark, 2013) in a manner similar to what is known for the MHC-II genes conferring adaptive immunity in other vertebrates. Such an extensive polymorphism of antimicrobial peptides has not been previously described in fish. Here, dynamic natural selection at hotspots of individual primary defenses may indicate the added importance of innate immunity when adaptive immunity is lacking.

Supplemental Information

Supplemental Information 1 File S1

All the supplemental figures in one file.

Click here for additional data file.

Figure S1 Alignment of amino acid sequences of exons 1, 2, and 3 combined, the conserved part of cathelicidin, from clones of various individuals of Atlantic cod and four closely related taxa

Highly polymorphic sites are boxed. The four conserved cysteine residues characterizing cathelicidin are shaded. Up arrows represent positively selected sites and down arrows negatively selected sites from Tables 1 and 2. Bsa.Gre (Boreogadus saida), Gch.Pac (Gadus chalcogrammus), Gma.Pac (Gadus macrocephalus), Gog.Gre (Gadus ogac) and Gmo (Gadus morhua) from various locations; Iceland (Gmo.Ice), Greenland (Gmo.Gre), Barents Sea (Gmo.Bar), Celtic Sea (Gmo.Cel), Baltic Sea (Gmo.Bal), Norway (Gmo.Nor), Faeroe Islands (Gmo.Far), Canada (Gmo.Can).

Click here for additional data file.

Figure S2 Alignment of nucleotide sequences of cathelicidin among clones from various individuals of Atlantic cod and four closely related taxa

Bsa.Gre (Boreogadus saida), Gch.Pac (Gadus chalcogrammus), Gma.Pac (Gadus macrocephalus), Gog.Gre (Gadus ogac) and Gmo (Gadus morhua) from various locations; Iceland (Gmo.Ice), Greenland (Gmo.Gre), Barents Sea (Gmo.Bar), Celtic Sea (Gmo.Cel), Baltic Sea (Gmo.Bal), Norway (Gmo.Nor), Faeroe Islands (Gmo.Far), Canada (Gmo.Can).

Click here for additional data file.

Figure S3 Maximum likelihood phylogenetic tree of Cathelicidin amino acid sequences of the conserved part

Phylogenetic tree built on amino acid sequences in exons 1, 2, and 3 combined, the conserved part of cathelicidin, of clones from various individuals of Atlantic cod and four closely related taxa. Bsa.Gre (Boreogadus saida), Gch.Pac (Gadus chalcogrammus), Gma.Pac (Gadus macrocephalus), Gog.Gre (Gadus ogac) and Gmo (Gadus morhua) from various locations; Iceland (Gmo.Ice), Greenland (Gmo.Gre), Barents Sea (Gmo.Bar), Celtic Sea (Gmo.Cel), Baltic Sea (Gmo.Bal), Norway (Gmo.Nor), Faeroe Islands (Gmo.Far), Canada (Gmo.Can).

Click here for additional data file.

Figure S4 Maximum likelihood phylogenetic tree of nucleotide sequences

Phylogenetic tree of nucleotide sequences of the cathelicidin gene from 43 representative clones of various individuals of Atlantic cod and four sister taxa. Bsa.Gre (Boreogadus saida), Gch.Pac (Gadus chalcogrammus), Gma.Pac (Gadus macrocephalus), Gog.Gre (Gadus ogac) and Gmo (Gadus morhua) from various locations; Iceland (Gmo.Ice), Greenland (Gmo.Gre), Barents Sea (Gmo.Bar), Celtic Sea (Gmo.Cel), Baltic Sea (Gmo.Bal), Norway (Gmo.Nor), Faeroe Islands (Gmo.Far), Canada (Gmo.Can).

Click here for additional data file.

Figure S5 Sliding window analysis of nucleotide diversity π and the scaled mutation rate θ for Cath1

Window length was 100 bp with a 25 bp step size.

Click here for additional data file.

Figure S6 Sliding window Tajima’s D for Cath1

Window length was 100 bp with a 25 bp step size.

Click here for additional data file.

Figure S7 Sliding window analysis of nucleotide diversity π and the scaled mutation rate θ for Cath3

Window length was 100 bp with a 25 bp step size.

Click here for additional data file.

Figure S8 Sliding window Tajima’s D for Cath3

Window length was 100 bp with a 25 bp step size.

Click here for additional data file.

We thank Jarle Mork (Norwegian University of Science and Technology), Kristján Kristjánsson (Marine Research Institute in Reykjavik), Grant Pogson (University of California at Santa Cruz), Remment ter Hofstede (Institute for Marine Resources and Ecosystem Studies in the Netherlands), and Michael Canino (National Oceanic and Atmospheric Administration) for help in securing some of the samples. We thank Valerie H. Maier and Arnar Pálsson for their valuable comments on the manuscript. We thank RC Lewontin for office space and encouragement and stimulating environment and discusion while writing the manuscript.

Additional Information and Declarations

Competing Interests

Author Contributions

Animal Ethics

DNA Deposition

The authors declare there are no competing interests.

Katrín Halldórsdóttir and Einar Árnason conceived and designed the experiments, performed the experiments, analyzed the data, contributed reagents/materials/analysis tools, wrote the paper, prepared figures and/or tables, reviewed drafts of the paper.

The following information was supplied relating to ethical approvals (i.e., approving body and any reference numbers):

The Icelandic Committee for Welfare of Experimental Animals, Chief Veterinary Office at the Ministry of Agriculture, Reykjavik, Iceland has determined that the research conducted here is not subject to the laws concerning the Welfare of Experimental Animals. (The Icelandic Law on Animal Protection, Law 15/1994, last updated with Law 157/2012.) DNA was isolated from tissue taken from dead fish on board research vessels. Fish were collected during the yearly surveys of the Icelandic Marine Research Institute. All research plans and sampling of fish, including the ones for the current project, have been evaluated and approved by the Marine Research Institute Board of Directors. The Board comprises the Director General, Deputy Directors for Science and Finance and heads of the Marine Environment Section, the Marine Resources Section, and the Fisheries Advisory Section. Samples were also obtained from dead fish from marine research institutes in Norway, the Netherlands, Canada and the US that were similarly approved by the respective ethics boards. The samples from the US used in this study have been described in Cunningham et al. (2009) and the samples from Norway in Árnason & Pálsson (1996). The samples from Canada consisted of DNA isolated from the samples described in Pogson (2001). The samples from the Netherlands were obtained from the Beam-Trawl-Survey (http://www.wageningenur.nl/en/Expertise-Services/Research-Institutes/imares/Weblogs/Beam-Trawl-Survey.htm) of the Institute for Marine Resources & Ecosystem Studies (IMARES), Wageningen University, the Netherlands, which is approved by the IMARES Animal Care Committee and IMARES Board of Directors.

The following information was supplied regarding the deposition of DNA sequences:

New sequences generated in this study have GenBank accession numbers KJ831349–KJ831391.

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
