# Peer review of "Trans-species polymorphism at antimicrobial innate immunity cathelicidin genes of Atlantic cod and related species"

_PeerJ, doi:10.7717/peerj.976_

## Round 0.1 · original submission · Major Revisions

This article is interesting and mostly comprehensive in describing phylogeny and patterns of selection in Cathelicidin genes of the representative ray-finned fish, Atlantic cod. However, addressing comments of the reviewers will help in improvement of this manuscript. Hence, we encourage authors to reply point by point comments of the reviewers with all possible improvements.

·

Basic reporting

This article is good designed with balanced experimental work. I recommend to accept this paper with revisions.

Authors should try to improve basic manuscript, at many places it is written word thesis, I think it should be hypothesis (e.g at page 9 line number 300-1, "If that were the case it would also strengthen our thesis of trans-species polymorphism because both sub clusters of Cath3 cluster by alleles and not by species." again line 437-39).

Cathelicidins are well known in mammals and even this gene is already studied in cathelicidin-knockout (Cnlp−/−) mice by Cho's group in 2007.

I recommend for revision this Manuscript.

Experimental design

designed with nice reporting but,

1. Authors need to better explanation of collection of sample.

2. Samples compared in this are collected by different methods and different organs of "Cod" hence variations could be due to these, authors need an explain for this part more thoroughly.

Validity of the findings

acceptable

Reviewer 2 ·

Basic reporting

See general comments below

Experimental design

See general comments below

Validity of the findings

See general comments below

Additional comments

This manuscript investigates phylogeny and patterns of selection in Cathelicidin genes of the Atlantic cod (and related species). The authors find multiple gene variants per individual and sequence clustering that suggests the presence of at least three rather than two paralogs, as previously assumed. These genes have antimicrobial function and are important for the innate immune system. The authors thus propose that they are evolving under balancing selection, potentially compensating for the apparent loss of MHC class II genes in the cod. Gene trees suggest some extent of trans-species polymorphis that would support the idea of balancing selection acting on these genes. And further analysis of sequence evolution points in the same direction.
This is an interesting study and the data analyses are comprehensive and mostly sound. Investigating the evolution of innate immune genes is very relevant for evolutionary ecologists as well as for population geneticists and fisheries sciences. However, there are a number of issues that I think need to be improved before the manuscript can be published. Below I list these issues in detail.


Majore issues:

More information is needed about the PCR amplification protocol. The cited company manual does not give a specific number of cycles, however this is critical for artifact formation during the amplification of polymorphic loci (e.g. Lenz & Becker 2008 Gene).

Was the primer CodCathR4 obtained from previous work (if so, please cite) or developed for this project (if so, please provide more information on the primer development/verification)?

The authors base a large part of their interpretation of balancing selection on the observation of trans-species polymorphism, but it seems, at least to me, that they are mixing two phenomena: Across the three (or more) paralogs, sequence variants cluster by paralog and not by species. This is obviously expected and does NOT represent trans-species polymorphism nor balancing selection. On the other hand, there seem to be some instances of real trans-species polymorphism WITHIN paralogs and this could be evidence of trans-species polymorphism. However, in the text these two different phenomena are not clearly differentiated and it thus appears as if the authors are mis-interpreting at least some of the results.
This distinction needs to be made more clear in the text and the interpretation adjuste accordingly.

The evidence for positive (balancing) selection from codon-based analyses is much weaker than suggested by the authors. For some unexplained reason, they appear to use a significance threshold of p<0.2, rather thant the widely accepted threshold of p<0.05 (not to mention any multiple-test correction due to using 5 different detection methods on the same set of codons). Once that more realistic threshold is applied, the number of codons that show any evidence of positive selection decreases to four (NOT 5!).

In this light, I think the authors need to tone down their interpretation of balancing selection quite a bit, and also be more precise in what they see as evidence and what is not.
It would also be nice to read a bit more in the discussion about other potential MHC II-loss compensating mechanisms in cod, such as the expansion of the MHC class I and TLR families, as described in the initial genome paper.


Minor issues:

L46-48: Primates are mammals, so please rephrase.

Please find a more logical color coding for the different localities/species in figure 1. Preferably, all Atlantic cod locations would be one color and the others have different colors.

Please correct Goc to Gog in legend of figure 1.

Fig 2 + 3: It is not clear where the bootstrap values belong, i.e. which nodes they are supporting. Please be more precise in locating them.

---

## Round 0.2 · accepted · Accept

Your manuscript is acceptable now. Congratulations!!!. However, please write in material and methods about organ or tissue source of the DNA, as suggested by one of the reviewer.

·

Basic reporting

I recommend minor revisions to accept this manuscript.

As detailed in my previous review, Authors still failed to outline clearly and to mention in manuscript about the samples compared, such as from which organ, tissue they were collected, authors still need an explain clearly about this part. Mention in the manuscript about the sample source (tissue etc.)

Experimental design

-

Validity of the findings

-

Additional comments

-

Reviewer 2 ·

Basic reporting

Good

Experimental design

Good

Validity of the findings

Good

Additional comments

I thank the authors for taking my comments into consideration. They have addressed all concerns to my satisfaction. I have no further issues with the submitted manuscript.